# Characterization of Activated Carbon Paper Electrodes Prepared by Rice Husk-Isolated Cellulose Fibers for Supercapacitor Applications

**DOI:** 10.3390/molecules25173951

**Published:** 2020-08-29

**Authors:** Hong Gun Kim, Yong-Sun Kim, Lee Ku Kwac, Hye Kyoung Shin

**Affiliations:** Institute of Carbon Technology, Jeonju University, 303 Cheonjam-ro, Wansan-gu, Jeonju-si, Jeollabuk-do 55069, Korea; hgkim@jj.ac.kr (H.G.K.); wva223g6@naver.com (Y.-S.K.); kwack29@jj.ac.kr (L.K.K.)

**Keywords:** supercapacitor, rice husk, cellulose fibers, activated carbon paper

## Abstract

For the preparation of activated carbon papers (APCs) as supercapacitor electrodes, impurity substances were removed from rice husks, before carbonization and various activation temperature treatments, to optimize electro chemical efficiency. The porosities and electrochemical performances of the ACPs depended strongly on activation temperature: The specific surface area increased from 202.92 (500 °C) to 2158.48 m^2^ g^−1^ (1100 °C). XRD and Raman analyses revealed that ACP graphitization also increased with the activation temperature. For activation at 1100 °C, the maximum specific capacitance was 255 F g^−1^, and over 92% of its capacitance was retained after 2000 cycles.

## 1. Introduction

In order to avoid further increases in the use of fossil fuels, minimizing energy loss is an important priority, and therefore the development of new eco-friendly and high-efficiency energy storage devices is required [1,2,3]. The types of supercapacitors categorized as electric double-layer capacitors (EDLCs) have been a focus of research and development because of their high-power densities, fast charge-discharge rates, low environmental pollution, and great cycle performances [4,5,6]. The performance of EDLCs is governed by the structure of their electrodes, the type of electrolyte, and the interaction between electrodes and electrolyte [7,8,9,10]. The primary mechanism for the functioning of EDLCs depends on the pore walls of their materials. High porosity, and therefore a large surface area, is required to increase charge storage [11,12,13,14,15,16].

Activated carbon materials play important roles as electrode materials and have been used to manufacture EDLCs, including commercially available commercial devices. This is because of the excellent electrochemical performance, resulting from their large surface area, tunable pore sizes, chemical resistance, non-corrosion properties, and good electrical conductivities [17,18,19,20,21,22,23,24]. However, the precursors of most carbon materials are usually derived from fossil fuels, which strongly detracts from the environmental benefits of their use. The growth of green and renewable energy-source materials is, therefore, required.

Rice husks, which are also called “rice hulls”, are the outer shells that cover rice kernels. These materials are one of the agricultural byproducts of rice processing. They are used as a fuel, feedstuff, and fertilizer in some countries, and are burnt or dumped as waste. In some cases, the burnt and dumped rice husks have become the cause of environmental pollution [25,26,27,28]. However, rice husks may be one of the most widely available agricultural materials. Recently, rice husks have been studied in supercapacitor applications as a porous carbon obtained via an activation process.

However, thus far, most reseacrh has been based on the preparation of a porous carbon powder, obtained by a carbon activation process after carbonizing the whole rice husk. Efforts to separate the husk according to its composition, such as cellulose, lignin, and hemicellulose, and then study its supercapacitance properties have not been undertaken [29,30,31,32,33,34].

Cellulose, lignin, hemicellulose, ash, and other substance in plant-based materials have different characteristics. Lignin and hemicellulose have an amorphous structure and act as the glue between cellulose fibers, being smoother than cellulose. However, cellulose is a semi-crystalline linear polymer that has excellent mechanical properties because of the high aspect ratio of its fibers [35,36,37,38].

In addition, hydroxyl groups existing on the surface of cellulose fibers can form hydrogen bonds with the neighboring cellulose fiber surface hydroxyl groups if the inter-fiber spacing is sufficiently low [39,40]. These merits of extracted cellulose fibers are taken advantage of in a variety of applications, such as paper, composites, coatings, etc.

In this study, we isolated cellulose fibers from rice husks as a carbon electrode precursor from which lignin, hemicellulose, ash, and other substances had been removed. Wood, coconut, and other natural-product-based activated carbon materials have been widely applied in EDLC. However, if these materials include many impurities, their use has several drawbacks, and when acting as EDLC materials, they can induce gas generation or a reduction in capacity during charge-discharge processes as electrodes, even if the materials are treated via carbonization and activation processing at high temperature. It is thought that activated carbon materials obtained from cellulose fibers from which lignin, hemicellulose, ash, and other substances have been removed might contribute to improving electrode performance. To promote the growth of supercapacitor electrode application and provision of new application possibilities, we adopted a new material processing approach in this study. Instead of investigating porous rice husk carbon powders that have not undergone the removal of lignin and hemicellulose, which has been the focus of recent research, we extracted bleached pulps from rice husks and prepared cellulose papers using only thermal pressing, without a binder. The obtained cellulose papers were carbonized to produce activated carbon papers with various pore sizes and properties that were dependent on the temperature of their activation after KOH treatment. The activated carbon papers (ACPs) were characterized by scanning electron microscope (SEM), the Brunauer-Emmett-Teller (BET), X-ray diffraction (XRD), and Raman spectral analyses, and then testing to assess their potential as EDLC materials.

## 2. Results and Discussion

### 2.1. Dependence of ACP Morphological Properties and Crystal Structure on Activation Temperatures

The SEM images in Figure 1 show the morphological characterization of the porous structure of the ACPs. A dependence of the morphology on activation temperatures is apparent. The surface of the carbon paper shown in Figure 1a has no porosity. After activation at different temperatures, we can see that the porosity of the ACP increases, and different sizes of porous structures are generated at different temperatures. Overall, there is a trend of increasing magnitude of porosity, as well as greater diversity among the porous structures, as the activation temperature increases. In particular, the surface morphology of ACP-1100 includes the richest variation and density of pores and grain-like particles (Figure 1e); however, the structures of the ACP fibers had diameters 10 µm less than those of the bleached rice husk cellulose fibers (Figure 1b) and retained shape stability during the activation process at this temperature (Figure 1f,g). These results, concerning the formation of well-developed pores under high-temperature activation, are expected, due to the fact that activation temperature elevation causes greater decomposition of carbon papers impregnated with potassium salts [41,42]. The rough surfaces and porous morphologies generated by such treatments can be helpful for increasing the degree of contact between the ACP electrode and the electrolyte.

The microstructural characterization of the ACPs was executed using N_2_ adsorption-desorption isotherms. In Figure 2, the volume absorption of N_2_ absorption on the ACP surfaces increased with activation temperature. In addition, for each of the different ACP surfaces, the absorbed volume increased steeply at a relative pressure (*p/p_0_*) < 0.1, corresponding to types I isotherms according to BET classification. This behavior is typical of microporous carbon materials. In Figure 2a, although the adsorption isotherms of all the ACPs resemble one another, the N_2_ absorbed volumes for ACP-500 and ACP-700 are low, whereas those for ACP-900 and ACP-1100 are high because of their large micro-porosity, as shown in the SEM images (Figure 1). To obtain a more detailed description of the pore sizes, pore-size distributions were computed. Figure 2b displays the micro-pore size distribution for all the ACP samples. The ACP samples are composed of micropore structures with sizes < 2 nm. However, in Figure 2b, ACP-700, ACP-900, and ACP-1100 display an ultra-micro-pore structure, i.e., pores with sizes below 0.7 nm. This is most obvious in the case of ACP-1100, for which the micro-pore distribution peaks at 0.33 nm, and subsequently peaks again at approximately 0.44 nm. These results indicate that the more extreme activation temperature had an important role in facilitating the formation of ultra micro-pores below 0.7 nm in size and micro-pore structures smaller than 2 nm in the ACP. Table 1 lists the pore characteristics for all the ACP samples. As listed in Table 1, the BET surface areas for ACP-500, ACP-700, ACP-900, and ACP-1100 showed 202.92, 751.05, 1569.32, and 2158.48 m^2^ g^−1^, respectively, and the proportion of the micro-pore volume relative to total volume is high at the activation temperature. For ACP-900 and ACP-1100, the higher specific surface area and abundance of ultra-micro and micro-pores are expected to be favorable towards electric capacity enhancement.

Figure 3a exhibits the XRD profiles of the ACP samples. All the ACPs have diffraction patterns that include two representative diffraction peaks at 2θ values close to 26° and 43°, associated with the (002) and (100) planes, respectively. As shown in Figure 3a, with increasing activation temperature, the width of the (002) peak decreases. The narrowing of this diffraction line with the activation temperature is ascribed to crystallization (increasing of atomic order) from a change in the degree of graphitization with the temperature of the second thermal treatment (after first carbonization, which was at 900 °C for all samples). ACP-1100, obtained by the highest-temperature thermal activation treatment, at 1100 °C, has the narrowest (002) peak, indicating that this sample underwent the greatest level of crystallinity development. This result is in accordance with our observations of the G-band in the Raman spectra. The Raman spectra in Figure 3b provide evidence of the microstructural conversion that the ACPs underwent during the activation, dependent on the activation temperature. This characterization is founded on the well-understood characteristic peaks at 1351 and 1610 cm^−1^, assigned to the D and G band, respectively, of graphite. The D band is associated with a disordered or defective graphite structure and the G band is related to the ordered-layered graphite structure, as it is assigned to intra-layer vibrations of sp^2^-bonded carbon atoms. Thus, the ratio of intensities ratios of these two bands, *I*_G_/*I*_D,_ can be used to estimate the degree of graphitization. As shown in Figure 3b, spectral changes at 1351 and 1610 cm^−1^ corresponding to the D and G bands follow a clear trend with the activation temperature. The *I*_G_/*I*_D_ intensity ratios are 1.21, 1.25, 1.33, and 2.94 for ACP-500, ACP-700, ACP-900, and ACP-1100, respectively. The ACP-1100 value is more than two times greater than those of all the other samples. These results indicate that the higher activation temperature increase the degree of graphitization, confirming our observations from the XRD data illustrated in Figure 3a.

### 2.2. Electrochemical Performance of the ACPs

The potential of the ACPs as electrode materials for supercapacitor applications was investigated using cyclic voltammetry (CV). The CV curves of the ACPs were measured within the potential range from −0.7 to 0 V at a potential scan rate of 50 mVs^−1^ (Figure 4a). At this scan rate, all ACF curves have rectangular shapes, suggesting nearly typical EDLC behavior. In addition, the rectangular area increased with the activation treatment temperature (ACP-1100>ACP-900 > ACP-700 > ACP-500). The increase in the rectangular area could be explained as reflecting an increase in the corresponding specific capacitance of the ACF electrode. Therefore, the ACP-1100 electrode, having the largest rectangular area, shows the greatest electrochemical capability as a result of enhanced permeation of the electrolyte because of its higher porosity and larger surface. Further, from the CVs for the ACP-1100 electrode acquired at different scan rates (Figure 4b), the rectangular area up to 100 mV s^−1^ increases without deformation as the scan rate increases, which constitutes an excellent rate performance.

Figure 5 displays the galvanostatic charge-discharge (GCD) results for the ACP-500, ACP-700, ACP-900, and ACP-1100 electrodes, and their evaluated specific capacitance values. All the samples were tested at a current density of 1 A g^−1^ (Figure 5a), as well as at various current densities from 1 to 5 Ag^−1^ (Figure 5b). In Figure 5a, the plots for all samples resemble in shape the typical triangular form that indicates good EDLC behaviors. The specific capacitance values of the ACP electrodes were calculated via the following expression:(1)$$\mathrm{Cs}\;=\;\frac{I△t}{m△V},$$
where I is the discharge current, Δt is the discharge time, m is the mass of the ACF electrode, and Δ*V* is the potential window.

At a current density of 1 A g^−1^, the specific capacitance of the ACP-500, ACP-700, ACP-900, and ACP-1100 electrodes are 110, 147, 200, and 255 F g^−1^, respectively. The capacitance value increases with the activation temperature for the ACPs, with ACP-1100 having the highest capacitance value. This increase in the capacitance is due to the increase in meso-porosity, which allows greater contact between the electrolyte ions and electrode and is also a result of the improvement of the electrical conductivity with the increase in graphitic content. The latter effect also occurs due to the increase in activation temperature, as mentioned above in the discussion of the BET and Raman analyses. The capacitance values of the ACPs obtained from cellulose fibers from which lignin, hemicellulose, ash, and other substances were removed to improve the electrode performance are comparable to those of the rice-husk derived activated carbon electrodes with the addition of a binder and high-conductive materials in other recent studies [43,44,45,46,47]. Thus, ACPs obtained from rice husk-extracted cellulose papers could be created as high-performance free-standing electrodes without the addition of a binder and conductive materials.

Figure 5b shows the specific capacitance of ACP electrodes at various current densities in the range of 1~5 A g^−1^. Specific capacitance values decreased as the current density increased. This is owing to the electrode resistance by the capacitance retention rate at a high current density [48].

To investigate the long-term cyclability of samples with the highest capacitance, ACP-1100, GCD tests were carried out at a current density of 5 A g^−1^ for 2000 cycles. As shown in Figure 6, the ACP-1100 electrode retains more than 92% of its capacitance after 2000 cycles, and this represents excellent specific capacitance retention.

### 2.3. Mechanical Performance

Figure 7a shows the stress–strain behaviors of the bleached pulp paper and ACPs. As shown in Figure 7a, the tensile strength of the ACPs was lower than that of the bleached pulp paper of the precursor and decreased with increasing the activation temperature. For ACP-1100, the tensile strength of the ACP-1100 decreased close to three times less than the bleached pulp paper. These results are due to the formation of micro pores, such as defects, in the fibers, as shown in Figure 1. However, in the photographs of Figure 7b,c, ACP-1100, with the lowest tensile strength and the highest porosity, maintained the paper shape and could bend despite its imperfect flexibility.

## 3. Materials and Methods

### 3.1. Materials

Rice husks are purchased from SAMWHA RICE MILL Co. (Hwaseong-si, Gyeonggi-do, Korea). All chemicals were analytical grade and used as received.

### 3.2. Preparation of the Activated Carbon Paper

First, rice husks were alkali-cooked and bleached using a two-step bleaching process (Table 2) to obtain cellulose fibers. The cellulose fibers were prepared as paper samples (thickness: 0.43 ± 0.051 mm, diameter: 350 ± 0.18 mm) using only the thermocompression method without a binder (Figure 8). Table 3 lists the chemical composition of raw rice husks and bleached rice husk pulps. Their contents were determined using the Technical Association of Pulp and Paper Industry (TAPPI) method, and the silica ash content was determined using the thermogravimetric analysis (TGA, TA, SDT 650, New Castle, DE, USA).

The cellulose papers were carbonized by heating at 900 °C for 1 h under an atmosphere of pure N_2_ (99.999%). To prepare the ACPs, the carbon papers were immersed with KOH solution (25 *wt*.%) and then dried at 80 °C. A second carbonization process was then performed at 500, 700, 900, and 1100 °C, respectively, for 1 h, under nitrogen atmosphere to complete the activation. The ACPs were subsequently washed with H_2_SO_4_ (0.1 M) to remove surplus KOH and neutralize the surfaces using distilled water before they were dried at 80 °C. The ACP samples were labeled ACP-500, ACP-700, ACP-900, and ACP-1100 according to their activation temperature condition. Figure 9 shows a schematic diagram illustrating the preparation of the ACP from rice husks.

### 3.3. Characterization

The surfaces of the ACPs were observed using SEM (HITACHI SU-70, Tokyo, Japan) at 200,000× and 5,000× magnifications. The specific area was calculated using BET analysis, and the total pore volume was detected from N_2_ adsorption data at a relative pressure (*p/p_0_*) of 0.99. Non-local density functional theory (NLDFT) was employed to evaluate the pore size distributions. The crystallinity of the ACPs was determined by XRD (RIGAKU, D/MAX-2500 instrument, Tokyo, Japan), with CuKα radiation operating at 40 kV and 30 mA. Raman spectra were acquired using an ARAMIS instrument (Horiba Jobin Yvon, Tokyo, Japan); a 514-nm laser was used, and the spectral range was from 500 to 3500 cm^−1^. Tensile testing was carried out with the ACPs. The ACP test samples with a width of 15 mm and length of 50 mm were measured using an Instron 5050 tester (Instron USA, Norwood, MA, USA) with a load cell of 1 kN.

### 3.4. Electrochemical Measurements

The electrochemical performances of the ACP electrodes were investigated by CV and galvanostatic charge tests using a CHI 660E electrochemical workstation (CH Instruments, Inc., Beijing, China). The samples, which consisted of only the ACP without the addition of binder or conductive materials, were tested using a three-electrode system with KOH solution (1 M) as the electrolyte. The ACP electrode was used as the working electrode, an Ag/AgCl electrode was employed as the reference electrode, and a platinum coil acted as the counter electrode. CV curves were measured in the potential range from −0.7 to 0 V using potential scan rates in the range of 10–100 mV s^−1^ and a step size 0.5 mV. GCD tests were undertaken over a step-wise increase in the current density between 1 and 5 A g^−1^ with a voltage range from −0.7 to 0 V vs. Ag/AgCl.

## 4. Conclusions

Plant resources, such as rice husks, are easy to obtain and widely applied as carbon electrode materials. We prepared bleached rice husk papers as a carbon electrode precursor in which lignin, hemicellulose, ash, and other substances in rice husks had been removed. Via carbonization and activation temperature variation, ACP electrodes with various porosities were obtained. The activation temperature exerted a strong influence on the porosity, crystal structure, and electrochemical performances of the ACPs. A higher activation temperature resulted in greater micro-pore volumes and increased graphitization. Among our results, the sample ACP-1100, for which the activation temperature was the highest, had the highest capacitance values and a maximum specific capacitance of 255 F g^−1^.

## Figures and Tables

**Figure 1 molecules-25-03951-f001:**
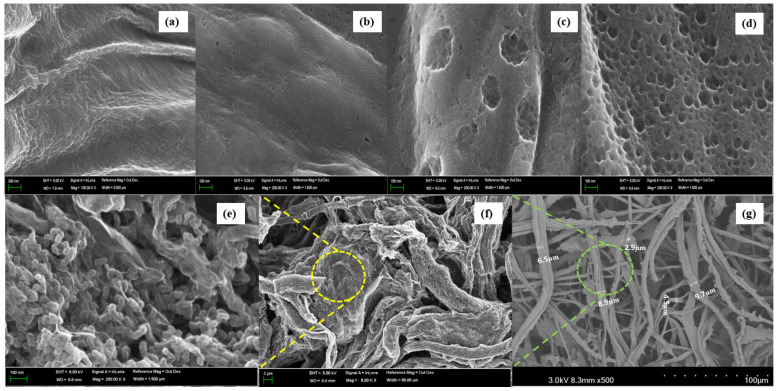
Morphological characterization of the activated carbon papers (ACPs) and SEM images of (**a**) carbon paper, (**b**) ACP-500, (**c**) ACP-700, (**d**) ACP-900, (**e**) (magnification: 20,000×), (**f**) (magnification: 5000×), and (**g**) (magnification: 500×) ACP-1100.

**Figure 2 molecules-25-03951-f002:**
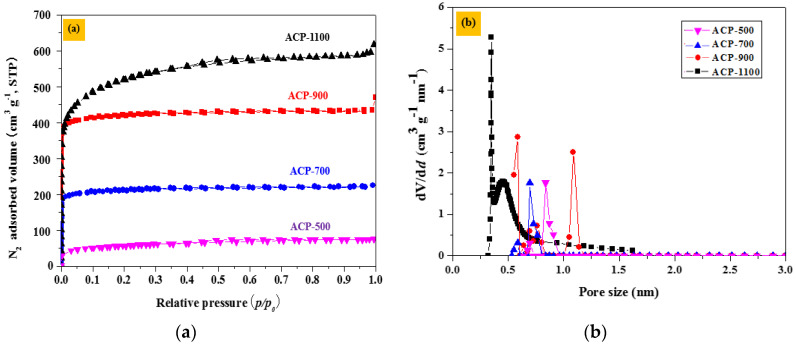
ACP N_2_ adsorption and micro porosity. (**a**) Experimentally measured adsorption-desorption isotherms and (**b**) computed micro-pore size distributions.

**Figure 3 molecules-25-03951-f003:**
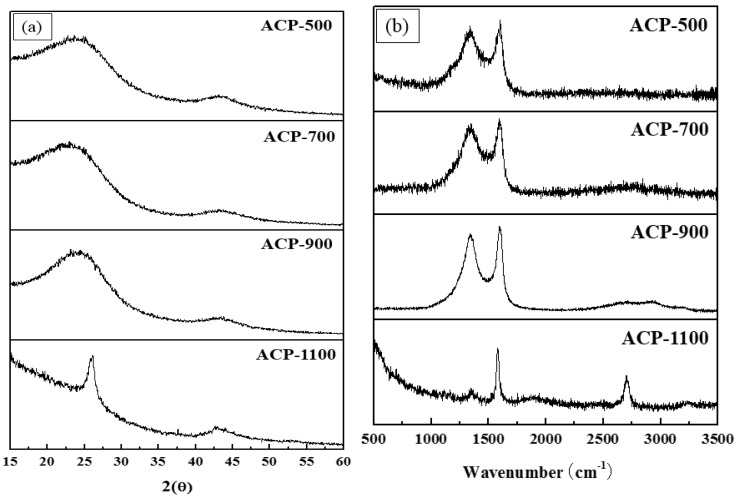
Crystal structure characterization: (**a**) XRD patterns and (**b**) Raman spectra of the ACPs.

**Figure 4 molecules-25-03951-f004:**
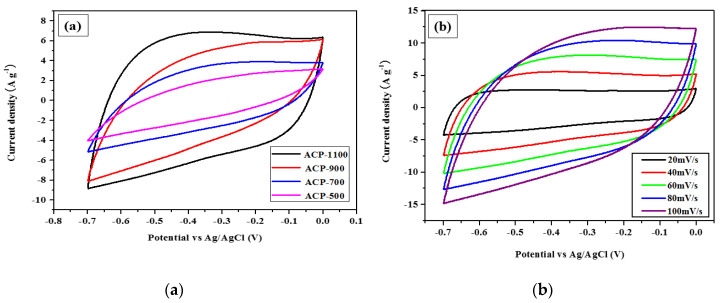
Electrochemical performance in 1 mol L^−1^ KOH: (**a**) cyclic voltammograms of ACP-500, ACP-700, ACP-900, and ACP-1100 electrodes, scan rate: 50 mVs^−1^ and (**b**) cyclic voltammograms of ACP-1100 electrode, with various scan rates between 10 and 100 mVs^−1^.

**Figure 5 molecules-25-03951-f005:**
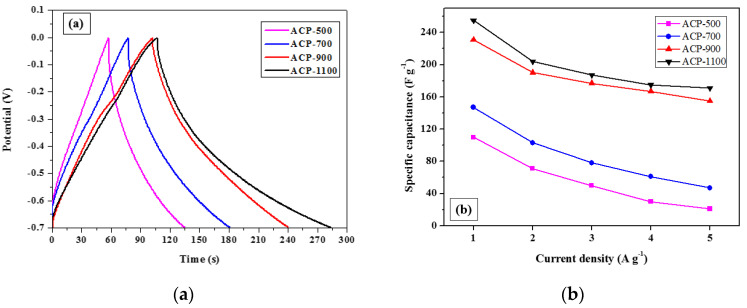
Galvanostatic charge-discharge (GCD) results: (**a**) GCD curves for ACP-500, ACP-700, ACP-900, and ACP-1100 electrodes with a current density of 1 A g^−1^ and (**b**) specific capacitance for each ACP electrode vs. current density.

**Figure 6 molecules-25-03951-f006:**
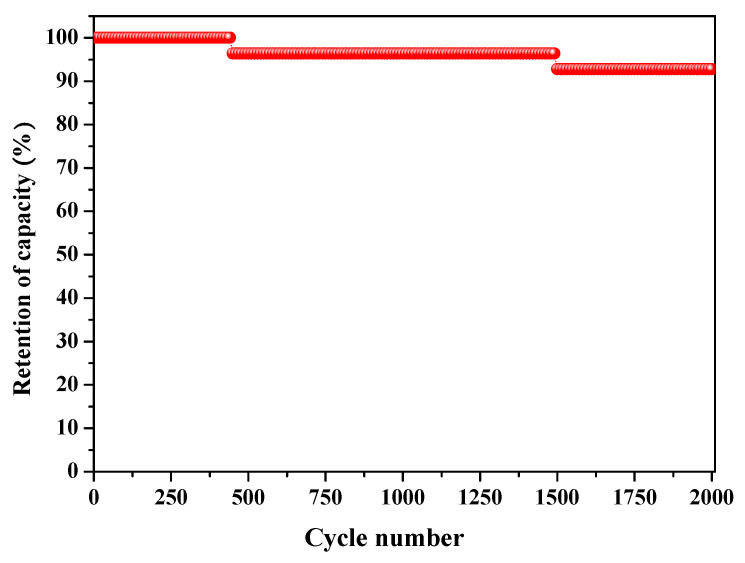
Electrochemical cyclic stability analysis of ACP-1100 at a current density of 5 A g^−**1**^.

**Figure 7 molecules-25-03951-f007:**
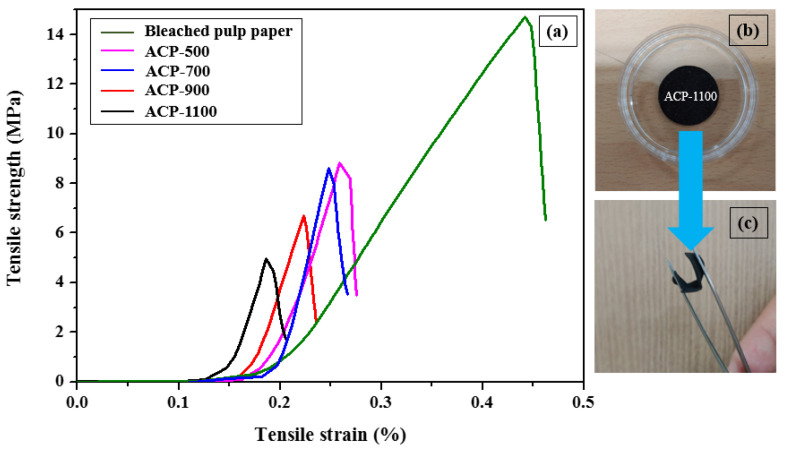
(**a**) Tensile stress–strain curves of the bleached pulp paper of the precursor and ACPs, and photographs of (**b**) ACP-1100 and (**c**) the bent shape of ACP-1100.

**Figure 8 molecules-25-03951-f008:**
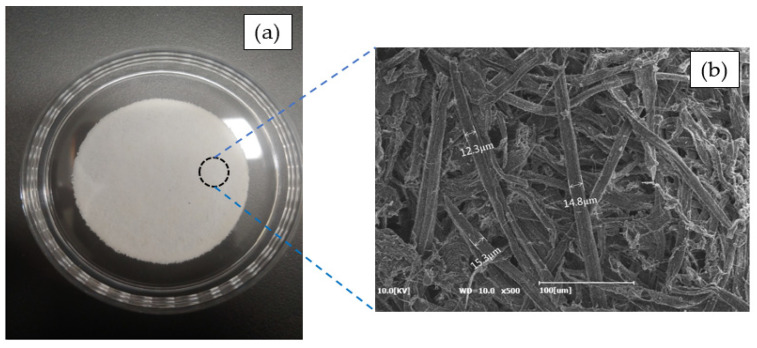
(**a**) Photographs and (**b**) SEM image of the bleached rice husk paper.

**Figure 9 molecules-25-03951-f009:**
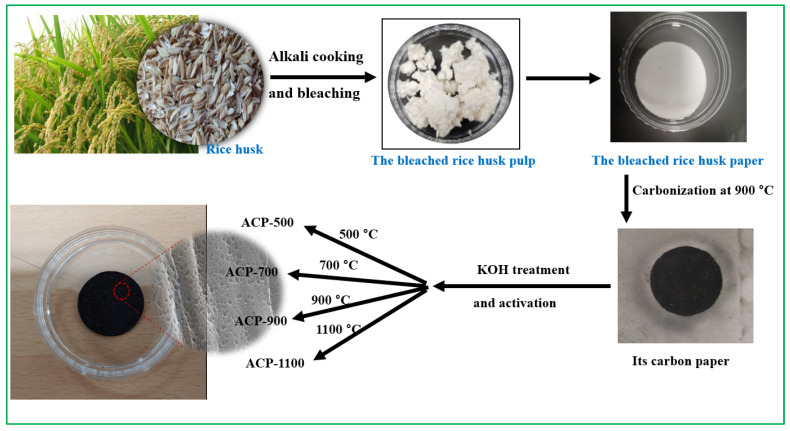
ACP preparation method: schematic diagram showing preparation of ACP-500, ACP-700, ACP-900, and ACP-1100 with photographs of the samples acquired during processing.

**Table 1 molecules-25-03951-t001:** Pore properties of ACP samples calculated from N_2_ adsorption measurement.

Samples	*S* _BET_	*V* _total_	*V* _micro_	*V* _meso_	*V*_micro_/*V*_total_(%)
ACP-500	202.924	0.116	0.091	0.025	61.54
ACP-700	751.05	0.261	0.256	0.00556	98.084
ACP-900	1569.32	0.789	0.638	0.151	80.862
ACP-1100	2158.48	0.939	0.884	0.055	94.143

**Table 2 molecules-25-03951-t002:** Alkali cooking and a two-step bleaching process for the isolation of cellulose.

	Alkali Cooking	1st Bleaching	2nd Bleaching
Chemicals	12 wt.% sodium hydroxide	2 wt.% sodium chlorite3 wt.% acetic acid	1.2 wt.% sodium hypochlorite
Temperature	120 °C	70 °C	Room temperature
Time	120 min	90 min	60 min

**Table 3 molecules-25-03951-t003:** Chemical composition of raw rice husks and bleached rice husk pulps.

	Cellulose (%)	Hemicellulose (%)	Lignin (%)	Silica Ash (%)
Raw rice husks	41.51 ± 1.32	28.12 ± 0.84	21.98 ± 0.61	20.44 ± 0.79
Bleached rice husk pulps	94.12 ± 2.40

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
