# Peer review of "Characterization of Activated Carbon Paper Electrodes Prepared by Rice Husk-Isolated Cellulose Fibers for Supercapacitor Applications"

_molecules, 2020, doi:10.3390/molecules25173951_

Round 1

Reviewer 1 Report

Dear Authors:

In my viewpoint, the manuscript can be accepted to publication after minor revision. In this sense, seems that organization of manuscript is not correct. Further amount of data and its discussion is presented prior to Materials and Methods presentation.

Also, I suggest doesn’t use the third person in the manuscript, this procedure sound as a testimonial, instead scientific sound. Then, at several parts of manuscript is finding ..."we...". My first suggestion is use impersonal form. Authors should delete from manuscript' text all adverbs. See, further revision of English level is fundamental. In specific, some structures of idea report are erroneous, words are not representative of ideas. See as an example, in 115 line; "Figure 3(a) exhibits the XRD analysis...", as a matter of fact, the Fig 3 doesn't contain any analysis, in fact show X-ray diffraction profiles. Also, this characterization technique doesn't exhibit any peak, X-ray diffraction has a set of diffraction lines, and each line is indexed to a crystallographic plane. In the 116 line, ... "...of approximately 26°and 43°,", change to ...close to 26... I would like to comment that "the narrowing..." of diffraction line is ascribed to the crystallization (increasing of atomic order), despite of the "graphitization"...

In the 199 line, again another problem with English level, see "The obtained ACPs...", here the construction of phrase is erroneous.

Well, as a whole, kind of comments provides above are interesting to revision of all manuscript. 

Author Response

Dear reviewer

I thank the reviewer for their very constructive comments on this manuscript.

The following revisions were made.

Also, I suggest doesn’t use the third person in the manuscript, this procedure sound as a testimonial, instead scientific sound.

→ The third researcher paid a significant portion of the material and analysis costs for this experiment, and provided equipment to produce good results. He will give us some advice on the improvement of the electrical conductivity and mechanical strength of the ACP during experimenting with in the future, and He will also be a big part of the way forward for a better paper. So I included the third researcher in this paper.

Then, at several parts of manuscript is finding ..."we...". My first suggestion is use impersonal form. Authors should delete from manuscript' text all adverbs.

→ I deleted most of adverbs as your comments.

See, further revision of English level is fundamental. In specific, some structures of idea report are erroneous, words are not representative of ideas. See as an example, in 115 line; "Figure 3(a) exhibits the XRD analysis...", as a matter of fact, the Fig 3 doesn't contain any analysis, in fact show X-ray diffraction profiles.

→ I revised “XRD analysis “to “XRD profiles

Also, this characterization technique doesn't exhibit any peak, X-ray diffraction has a set of diffraction lines, and each line is indexed to a crystallographic plane. In the 116 line, ... "...of approximately 26°and 43°,", change to ...close to 26...

→ I revised “approximately “to “close to”

I would like to comment that "the narrowing..." of diffraction line is ascribed to the crystallization (increasing of atomic order), despite of the "graphitization"...

→ The following revision was implemented based on your comments

Before: The narrowing of this peak with activation temperature could result from a change in the degree of graphitization with the temperature of the second thermal treatment (after first carbonization, which was at for all samples 900 °C).

After: The narrowing of this diffraction line with activation temperature is ascribed to the crystallization (increasing of atomic order) from a change in the degree of graphitization with the temperature of the second thermal treatment (after first carbonization, which was at for all samples 900 °C).

In the 199 line, again another problem with English level, see "The obtained ACPs...", here the construction of phrase is erroneous.

→ Based on your comments, I revised “The obtained ACPs” into “The ACPs”.

Thank you again for your review of this paper

Best regards.

Hye Kyoung Shin

Reviewer 2 Report

This paper presents a carbon paper based on rice husk pulp fibers, aiming for electrochemical application. The preparation route, nanostructure and electrochemical performance are studied. The whole paper is clearly written and easy to follow. However, it is more like a showcase report which is missing lots of detailed information. Some new experiments are needed, so that major revision is recommended.

1. Instruction is fairly long for being a single paragraph. Any way to divided it into several paragraphs? Also, making activated carbon papers is not a new topic, while maybe making such papers based on rice husk pulp fibers is new (but please go through the literature carefully when making such statement). Please add more information about this.

2. Figure 3. “All the ACPs have diffraction patterns that include two representative diffraction peaks at 2θ values of approximately 26°and 43°, associated with the (002) and (100) planes, respectively.” 1) 002 and 100 planes for graphene. 2) unfortunately, the peak position for ACP-500, 700 and 900 is not at 26 degrees, it is at ~ 23, which is overlapping with cellulose 200 plane. That’s why they have a broad peak. For ACP-1110, it is highly possible that most the crystalline cellulose is gone, so that there is a sharp graphene 002 peak. Please rewrite the discussion about the Figure 3.

3. Chemical composition: This study is trying to remove all the non-cellulosic components in the first step, but there is no information supporting this, i.e. chemical composition change and yield after cooking and after bleaching. The residual lignin and hemicellulose and the crystallinity of the bleached fibers will affect the carbonization process for sure. And what about the yield?

4. This is a critical one: General structure of the paper (before and after carbonization): The SEM is not showing the morphology of paper structure. What are the length and width of individual fiber? What is the density and thickness of paper? What about the pores at microns when big fibers forming the network? What are the changes of all these after carbonizations?

5. This is a critical one: any characterizations on the mechanical properties of the carbonized papers? One of the biggest challenge for carbonized paper materials is they are fragile and not easy to handle. Any information on this?

6. This is a critical one: For electrochemical performance, any comparison between this study and other carbonized papers (Kraft wood papers, grass papers, etc). It is obvious that this material is going to compete against those materials instead of rice husk powders (in the introduction).

7. Figure 2 top image is cropped wrongly…. The image resolution of Figure 2, 3, 4 and 5 is too low – blurring figures….

Author Response

Dear reviewer

I thank the reviewer for their very constructive comments on this manuscript.

The following revision were made.

  1. Instruction is fairly long for being a single paragraph. Any way to divided it into several paragraphs? Also, making activated carbon papers is not a new topic, while maybe making such papers based on rice husk pulp fibers is new (but please go through the literature carefully when making such statement). Please add more information about this.

→ Based on your comments, I divided the introduction into several paragraphs.

→ Based on your comments, I tried to add more information. However, it seems that we are the first to manufacture a free-standing electrode obtained from rice husk pulp fibers which lignin, hemicellulose, and so on are removed. So, I didn’t add it because there isn’t any comparable information to compare.

  1. Figure 3. “All the ACPs have diffraction patterns that include two representative diffraction peaks at 2θ values of approximately 26°and 43°, associated with the (002) and (100) planes, respectively.” 1) 002 and 100 planes for graphene. 2) unfortunately, the peak position for ACP-500, 700 and 900 is not at 26 degrees, it is at ~ 23, which is overlapping with cellulose 200 plane. That’s why they have a broad peak. For ACP-1110, it is highly possible that most the crystalline cellulose is gone, so that there is a sharp graphene 002 peak. Please rewrite the discussion about the Figure 3.

→ The following revision was implemented based on your comments.

Before: The narrowing of this peak with activation temperature could result from a change in the degree of graphitization with the temperature of the second thermal treatment (after first carbonization, which was at for all samples 900 °C).

After: The narrowing of this diffraction line with the activation temperature is ascribed to the crystallization (increasing of atomic order) from a change in the degree of graphitization with the temperature of the second thermal treatment (after first carbonization, which was at 900 °C for all samples).

  1. Chemical composition: This study is trying to remove all the non-cellulosic components in the first step, but there is no information supporting this, i.e. chemical composition change and yield after cooking and after bleaching. The residual lignin and hemicellulose and the crystallinity of the bleached fibers will affect the carbonization process for sure. And what about the yield?

→ I added the chemical composition based on your comments

After: Table 3 lists the chemical composition of raw rice husks and bleached rice husk pulps. Their contents were determined using the Technical Association of Pulp and Paper Industry (TAPPI) method, and the silica ash content was determined using thermogravimetric analysis (TGA, TA, SDT 650, New Castle, DE, USA).

Table 3. Chemical composition of raw rice husks and bleached rice husk pulps

Cellulose (%)

Hemicellulose (%)

Lignin (%)

Silica Ash (%)

Raw rice husks

41.51 ± 1.32

28.12 ± 0.84

21.98 ± 0.61

20.44 ± 0.79

The bleached rice husk pulps

94.12 ± 2.40

  1. This is a critical one: General structure of the paper (before and after carbonization): The SEM is not showing the morphology of paper structure. What are the length and width of individual fiber? What is the density and thickness of paper? What about the pores at microns when big fibers forming the network? What are the changes of all these after carbonizations?

→ SEM images were added in Figure 8, and the following description was included: “The cellulose fibers were prepared as paper samples (thickness: 0.43 ± 0.051 mm, diameter: >10 µm) using only the thermocompression method without a binder (Figure 7).”

Figure 8. Photographs (a) and SEM image (b) of the bleached rice husk paper.

  1. This is a critical one: any characterizations on the mechanical properties of the carbonized papers? One of the biggest challenge for carbonized paper materials is they are fragile and not easy to handle. Any information on this?

→ The “Mechanical performance” section and Figure 7 were added.

After: Figure 7(a) shows the stress-strain behaviors of the bleached pulp paper and ACPs. As shown in Figure 7(a), the tensile strength of the ACPs was lower than that of the bleached pulp paper of the precursor and decreased with increasing the activation temperature. For ACP-1100, the tensile strength of the ACP-1100 decreased close to three times less than the bleached pulp paper. These results are due to the formation of micro pores, such as defects, in the fibers, as shown in Figure 1. However, in the photographs of Figure 7 (b) and (c), ACP-1100 with the lowest tensile strength and the highest porosity maintained the paper shape and could bend despite its imperfect flexibility.

Figure7. (a) Tensile stress-strain curves of the bleached pulp paper of the precursor and ACPs and photographs of ACP-1100.

  1. This is a critical one: For electrochemical performance, any comparison between this study and other carbonized papers (Kraft wood papers, grass papers, etc). It is obvious that this material is going to compete against those materials instead of rice husk powders (in the introduction).

→ The electrochemical performance condition of other studies is different for rice husk-based activated carbon materials and carbonized papers as electrodes. The capacitance values are higher or lower than those of the ACPs in this study because most of researchers added a binder and conductive materials. However, our ACPs are free-standing electrode without the addition of binder and conductive materials. We attempted to compare other carbonized paper or carbonize rice husk powders with the ACPs as electrodes; however, it was difficult to compare the results of other researchers. To apply your recommendation, the following sentence was added: “Thus, high performance free-standing rice husk-based ACPs could be created without the addition of a binder and conductive materials.”

  1. Figure 2 top image is cropped wrongly…. The image resolution of Figure 2, 3, 4 and 5 is too low – blurring figures…

→ The figures were revised based on your comments

Thank you again for your review of this paper

Best regards.

Hye Kyoung Shin

Round 2

Reviewer 2 Report

Great work on the revision! I would recommend this manuscript to be published with only two minor comments:

1. “The cellulose fibers were prepared as paper samples (thickness: 0.43 ± 0.051 mm, diameter: > 10 μm)” – According to Figure 7, “diameter” should not be 10 microns (too small). Maybe 10 mm or 10 cm?

2. “Thus, high performance free-standing rice husk-based ACPs could be created without the addition of a binder and conductive materials.” – Isn’t it one of the highlights in this manuscript? I would recommend to add a couple more sentences to explain what other ways to make similar materials such as using binders and additives (CNT, graphene and metal). Actually, this goes back to my previous comment 1 about the introduction. The present rice husk-based ACPs is going to compete against those paper materials with binders and additives. The advantage here might be cheap resource and easy preparation (author should know this better). Such comparison will help to improve the “Novelty”. 

Author Response

Dear reviewer

I thank the reviewer for their very constructive comments on this manuscript.

The following revision were made.

  1. “The cellulose fibers were prepared as paper samples (thickness: 0.43 ± 0.051 mm, diameter: > 10 μm)” – According to Figure 7, “diameter” should not be 10 microns (too small). Maybe 10 mm or 10 cm? 
  2. → I revised “diameter: > 10 µm to “diameter: 350 ± 0.18 mm” based on your comments
  3. “Thus, high performance free-standing rice husk-based ACPs could be created without the addition of a binder and conductive materials.” – Isn’t it one of the highlights in this manuscript? I would recommend to add a couple more sentences to explain what other ways to make similar materials such as using binders and additives (CNT, graphene and metal). Actually, this goes back to my previous comment 1 about the introduction. The present rice husk-based ACPs is going to compete against those paper materials with binders and additives. The advantage here might be cheap resource and easy preparation (author should know this better). Such comparison will help to improve the “Novelty”.
  4. → The following revision was added based on your comments.

After: The capacitance values of the ACPs obtained from cellulose fibers that lignin, hemicellulose, ash, and other substances were removed to improving electrode performance are comparable to those of the rice-husk derived activated carbon electrodes with the addition of a binder and high conductive materials in other recent studies [44-48]. Thus, ACPs obtained from rice husk extracted-cellulose papers could be created as high performance free-standing electrodes without the addition of a binder and conductive materials.

     → I also added the references related to this.

  1. Xue, B.; Wang, X.; Feng, Y.; Chen, Z.; Liu, X. Self-template synthesis of nitrogen doped porous carbon derived from rice husks for the fabrication of high volumetric performance supercapacitors. J. Energy Storage 2020, 30, 1-8.
  2. Yuan, C.; Lin, H.; Lu, H.; Xing, E.; Zhang, Y.; Xie, B. Synthesis of hierarchically porous MnO2/rice husks derived carbon composite as high-performance electrode material for supercapacitors. Appl. Energy 2016, 178, 260-268.
  3. He, X.; Ling, P.; Yu, M.; Wang, X.; Zhang, X.; Zheng, M. Rice husk-derived porous carbons with high capacitance by ZnCl2 activation for supercapacitors. Electrochim. Acta 2013, 105, 635-641.
  4. Tamilselvi, R.; Ramesh, M.; Lekshmi, G.S.; Bazaka, O.; Levchenko, I.; Bazaka, K.; Mandhakini, M. Graphene oxide-based supercapacitors from agricultural wastes: A step to mass production of highly efficient electrodes for electrical transportation systems. Renew. Energy 2020, 151, 731-739.
  5. Badhulika, S.; Gopalakrishnan, A. Ultrathin graphene-like 2D porous carbon nanosheets and its excellent capacitance retention for supercapacitor. J. Indus. Eng. Chem. 2018, 68, 257-266.

Thank you again for your review of this paper

Best regards.

Dr. Hye Kyoung Shin
